# The Rediscovery of Dynamic BPM: From Theoretical Margins to Mainstream Practice

Marek Szelągowski[1][0000-0002-5114-6793], Justyna Berniak-Woźny[1][0000-0002-3156-5755]
and Piotr Śliż[2][0000-0001-6776-3369]

[1] Systems Research Institute of the Polish Academy of Sciences, Warsaw, Poland
[2] University of Gdańsk, Poland
`piotr.sliz@ug.edu.pl`

**Abstract.** Dynamic Business Process Managment (dynamic BPM), developed since approximately 2005, is a case study in productive forgetting: a concept that was marginalised in BPM research not because it was wrong, but because it was misaligned with the dominant paradigm of predictability, ex ante standardisation, and workflow control. This article argues that dynamic BPM was, in fact, ahead of its time - and that contemporary BPM is quietly fulfilling its core logic without explicitly naming it. Drawing on a literature review spanning various BPM concepts, we identify three foundational assumptions of dynamic BPM that are now embedded in mainstream practice: recognition of the diverse nature of business processes, adaptation of process execution at runtime, and the pivotal role of knowledge, engagement, and responsible autonomy of the process participants. We further explain why the concept was pushed to the margins of academic discourse, and argue that the rise of hyperautomation and agentic AI makes its recovery both timely and necessary — though requiring extension to encompass AI governance, shared human–AI agency, and broader BPM objectives. Dynamic BPM was not a failed experiment. It was a forgotten framework that practice adopted and theory overlooked.

**Keywords:** dynamic BPM, semi-structured processes, knowledge management, adaptive case management, hyperautomation, agentic AI, human–AI agency, BPM governance.

## 1 Introduction

In science and management, some ideas do not disappear because they are wrong. They disappear because the field is looking elsewhere. Dynamic Business Process Management (dynamic BPM) is precisely such a case. Developed since approximately 2005, dynamic BPM was not an attempt at replacing traditional Business Process Management (BPM) but to extend it with a management logic suited to processes executed under conditions of variability, partial unpredictability, and the growing importance of practical knowledge arising during execution [1]. Its central departure was from the assumption that an organisation must know and impose ex ante the full, optimal course of a process — towards an approach in which process participants, acting within the

boundaries of their authorisations and constraints, co-shape the manner of achieving the goal in accordance with the execution context [1,2]. From its outset, dynamic BPM carried a dimension that was not only processual but also managerial: it emphasised the dynamism of process participants, their engagement, creativity, readiness to experiment, and responsible autonomy [1,3]. The concept did not gain traction in mainstream BPM research. Yet the organisational world it was designed to describe did not stand still. With the development of Industry 4.0 , the emerging Industry 5.0 and the increasingly widespread use of AI, real-time data, event-driven systems, and hyperautomation technologies, contemporary BPM is steadily moving away from focusing exclusively on fully predictable and repeatable processes [4–6]. Research on the nature of business processes confirms that organisations simultaneously operate across a spectrum: structured, semi-structured and unstructured processes [7–9]. A 2024 study by the Polish BPM Community found that fully structured, predictable processes account for only approximately 35% of all processes in contemporary organisations — and that semi-structured processes, which constitute the majority, are typically of high or very high importance [6]. BPM can no longer be equated solely with designing and enforcing a single optimal flow. It must encompass the management of variability, the execution context, and the knowledge that process participants bring to bear [9–11]. This is precisely the terrain that dynamic BPM had already mapped. Its marginalisation was not the result of conceptual weakness, but of misalignment with the dominant paradigm of standardisation, ex ante modelling, and workflow control — and with the successive waves of technology that reinforced that paradigm: workflow engines, document management, RPA, process mining [8,12,13]. Dynamic BPM remained on the periphery of academic discourse even as its assumptions were being quietly operationalised in practice by Adaptive Case Management [13,14]. Paradoxically, the current moment makes its recovery both possible and urgent. The extension of BPM to partially predictable and unpredictable processes, the deepening integration of BPM with Knowledge Management (KM), and the shift thanks to hyperautomation and AI of operational decisions closer to where real knowledge resides — all of these reflect assumptions that dynamic BPM articulated first [1,5,9,15]. And the rise of AI-augmented BPM and Agentic BPM signals that the next stage of evolution will demand something more still: a new account of agency, autonomy, and governance in environments where humans and artificial process actors work alongside one another [4,16].

This article pursues three objectives: to identify which foundational assumptions of dynamic BPM are already reflected in contemporary BPM; to explain why the concept was marginalised despite its relevance; and to determine what extensions it requires in the era of hyperautomation, AI, and agentic process participation. Dynamic BPM is treated here not as a historical curiosity, but as a forgotten framework that practice adopted and theory overlooked — and as a natural point of departure for its further evolution [1,16]. Three research questions structure the inquiry:

RQ1: Which foundational assumptions of dynamic BPM are reflected in contemporary BPM?

RQ2: Why and how was dynamic BPM marginalized in mainstream BPM discourse despite its importance in understanding the needs of modern business?

RQ3: What conceptual extensions are needed for dynamic BPM to remain relevant in the era of hyperautomation, AI, and agentic participation in business processes?

The article is composed of six parts. Following this introduction, Section 2 presents the methodological approach. Section 3 traces the evolution of BPM. Section 4 presents the concept of dynamic BPM, its essence, and its three foundational principles. Section 5 is discursive: it analyses the causes of dynamic BPM's marginalisation and identifies directions for its extension under conditions of hyperautomation and growing AI participation. Section 6 concludes.

## 2 Study Design

This article combines a literature review with interpretive synthesis. Rather than generating new empirical findings, it reconstructs and repositions an existing conceptual framework — dynamic BPM — in light of the trajectory of contemporary BPM research and practice. The analysis encompassed three interconnected literature streams: foundational and contemporary BPM research, including works on process typology and BPM evolution; research on the diverse nature of business processes and their relationship with Knowledge Management, particularly semi-structured and knowledge-intensive processes; and works on hyperautomation, AI-augmented BPM, and agentic process participation. Sources were selected purposively, prioritising conceptual and review publications that directly address process dynamics, the role of knowledge, human-centricity, and responsible autonomy — supplemented by selected empirical works where these provided direct evidential grounding.

The review was structured around the three research questions: RQ1 guided the identification of conceptual convergences between dynamic BPM and contemporary BPM; RQ2 oriented the analysis towards the intellectual and technological conditions shaping BPM discourse during dynamic BPM's marginalisation; RQ3 focused on identifying gaps where dynamic BPM requires updating. A targeted bibliometric analysis using Scopus (TITLE-ABS-KEY: "dynamic business process management" OR "dynamic BPM") was also conducted to assess the scale of academic engagement with the concept; results are reported in Section 5. The principal limitation is the conceptual nature of the study — the arguments are interpretive and not validated through primary empirical data. This is an acknowledged feature of the contribution: the article's aim is to reopen a conversation that was prematurely closed, not to close it definitively.

## 3 The Evolution of BPM

### 3.1 The Business Process

There are many definitions of a process [1,17,18], but at its core, a process is an ordered way of achieving a goal. In the spirit of Deming, it can be conceived as a system of interdependent activities directed towards an outcome whose effectiveness depends on managing the whole, not merely optimising individual parts [19]. So understood, a process need not be fully predictable or repeatable. In some cases the path

to the goal is known before action begins; in others it reveals itself during execution, shaped by context, available resources, the knowledge of the participants, and emerging events. This applies to sales, new solution design, patient treatment, complex service delivery — processes where value is created not only in the final result, but also in the knowledge verified or generated along the way, and in the tacit knowledge, skills, and competencies that participants develop through execution. The process is not only a vehicle for output; it is a space for learning, adaptation, and the creation of new knowledge. They focused on the dimension that was sufficient for mass production: predictability, repeatability, standardisation, and control. Under those conditions, such a narrowed concept was from their point of view entirely rational and sufficient. The problem arises when it is treated as universal and transposed onto processes whose course cannot be fully determined ex ante.

## 3.2 From Stability to Partial Predictability

The evolution of BPM was not limited to refining tools for modelling, optimising, monitoring, and automating processes. It gradually produced a paradigm shift: from predictability and the standardisation of repeatable processes towards acknowledgement that many organisational processes are only partially predictable and cannot be fully specified in advance [1,21,22]. This shift was also associated with an expansion of BPM's scope — from selected operational processes to the full spectrum of organisational activity. In such processes, the path to the goal cannot derive solely from a prior design prepared by management and experts; it must be co-created by process participants in accordance with the execution context, available resources, and the knowledge and skills they possess. It is here that contemporary BPM begins to reflect the fundamental assumptions of dynamic BPM. It extends process management beyond fully predictable processes to include semi-structured and unstructured ones. It recognises the necessity of runtime adaptation rather than adherence to a fixed template. And it increasingly acknowledges that process effectiveness depends not only on the quality of prior design but on the knowledge, engagement, and dynamism of participants — and on the organisation's capacity to learn from actual execution. These three directions of change were earlier articulated explicitly in dynamic BPM, and form the basis for its reinterpretation today. This evolution signals not only a broader scope for BPM but a partial return to a fuller understanding of the process as a way of achieving a goal within a specific context. The objective of contemporary BPM is becoming not merely the ordering and implementation of optimised processes but the creation of conditions in which the value of standardisation and operational efficiency combines with the value created by participants in the course of execution.

## 3.3 Business Processes in Contemporary Organisations

Recent research confirms that contemporary organisations simultaneously operate across a spectrum: structured processes, structured processes with ad hoc exceptions, unstructured processes with predefined fragments, and completely unstructured processes [7–9]. A 2024 study by the Polish BPM Community found that fully structured,

predictable processes account for only approximately 35% of processes in contemporary organisations — and that semi-structured processes, which constitute the majority, are typically of high or very high importance for organizations [6,22] (Fig. 1). Under conditions of hyperautomation, the value arising from knowledge created and operationalised during execution is growing further — manifesting both in process results and in the new tacit knowledge acquired by participants, whether human or artificial.

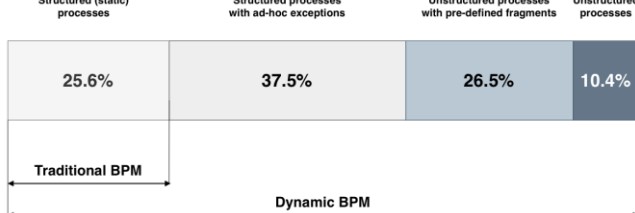

**Fig. 1.** Diversity of execution dynamics in business processes. *Source: Authors' own elaboration based on [6].*

Contemporary BPM must encompass those processes, whose effective execution requires the dynamism of participants and the operationalisation of knowledge management — both for operational efficiency and for continuous readiness to improve and innovate [15]. This evolution — the broadening of BPM's scope, the return to a fuller conception of the process, and the revaluation of dynamic processes — demonstrates that contemporary BPM is already fulfilling in practice the three fundamental assumptions of dynamic BPM: diverse process nature, runtime adaptation, and the co-determining role of participant knowledge, engagement, and dynamism.

## 4 The Concept of Dynamic BPM

The starting point for dynamic BPM was practical: the observation that increasingly sophisticated IT systems, deployed to support both operations and management, were delivering diminishing and increasingly disappointing returns [23]. The search for causes led to a diagnosis that traditional BPM's core method — imposing upon process participants a flow modelled and optimised ex ante, without accommodation for execution context — was systematically failing a growing share of organisational processes.

### 4.1 Identified Principal Weaknesses of Traditional BPM

The problem was not with traditional BPM as such, but with its scope of application. Treating every process as a mandatory sequence to be reproduced according to a predetermined template works well for fully structured, repeatable work. It proves deeply inadequate for semi-structured and unstructured processes, where outcomes depend significantly on the knowledge, experience, and situational judgement of process participants. In such processes, imposing the mode of execution suppresses initiative, limits the use of tacit knowledge, and obstructs the creation of new knowledge. It also

generates what management literature calls the "hidden factory" — informal workarounds that allow for achieving the goal of process execution and satisfy clients but remain invisible to the organisation, foreclosing systematic learning. The result is not only slower adaptation, but a gradual erosion of participant engagement: when the prescribed flow fails, responsibility shifts to whoever designed it.

## 4.2 The Essence of Dynamic BPM

Dynamic BPM was not, and is not, a proposal to replace traditional BPM or merely to increase process flexibility through technical means. It is not "dynamic processes" or "flexible processes." Dynamic BPM was an extension of traditional BPM philosophy adapting it to the VUCA world — one in which the capacity to rapidly use practical knowledge arising during process execution becomes a critical organisational capability. In this concept, understanding processes as flexible, agile, adaptable, contextual, or knowledge-intensive is not a goal, but a consequence of using, or rather executing, the dynamism of process executors from the need to implement business processes in accordance with an imposed model [1,21,24]. Among the various labels circulating in literature and practice — agile, adaptive, intelligent, cognitive, human — the term *dynamic* was chosen deliberately, to foreground the dynamism of process participants as the real source of organisational development potential. Knowledge, procedures, resources and IT systems matter; but to fully use them, the will to act, the engagement, the creativity, and the readiness to take responsible decisions in the course of work. It is upon this dynamism that improvements, limited experiments, and the generation of new knowledge depend. The essence of dynamic BPM is therefore to create organisational conditions thanks to which this dynamism can be fully expressed: within the limits of granted authorisations and constraints, process participants manage execution in accordance with established goals, execution context, available knowledge, skills, and resources. Thanks to this, it also enables ambidextrous BPM, as it allows for the verification and possible rejection of old, no longer useful knowledge, as well as the creation and capture of new knowledge during the implementation of processes, thus de facto creating a learning organization [24].

## 4.3 Principles of Dynamic BPM

Three principles operationalise this logic, each directly neutralising one of the weaknesses of traditional BPM identified above [1]:

**Principle I — Comprehensiveness and continuity.** Dynamic BPM must be oriented towards the organisation's strategic goals through the management of the entire value creation process, not isolated fragments. Individual areas, process groups, and single processes should consistently lead towards a common goal aligned with the organisational strategy — limiting the sub-optimisation risk in which one unit's gains worsen overall performance [25].

**Principle II — Evolutionary variability during execution.** For semi-structured processes — the majority, and typically the most strategically significant — decisions about execution must be made where real operational knowledge resides: with process

participants. Processes must be defined and managed so that, within granted authorisations and constraints, participants can supplement, modify, and sometimes reconfigure the process flow. This principle allows for the use of tools described in the literature as "dynamic" or "flexible" and methodologies named as "agile" or "contextual" as means subordinated to the main goal: creating value by using the dynamism of business process participants [20,26].

**Principle III — Execution is documentation.** Among the most important values created during process execution is knowledge. Dynamic BPM must therefore operate in an environment where executing a process is simultaneous with documenting it — enabling real-time comparison of standard and actual flows, identification of participant-introduced innovations, evaluation of their consequences, and continuous updating of organisational knowledge.

Together, these principles shift the decision-making architecture of the organisation. Choices about how work is performed move to where real knowledge of the client's situation, execution context, and process goal reside. This is empowerment in a precise sense: not unlimited freedom, but responsible autonomy within defined boundaries, coupled with an obligation to disclose the actual course of execution. The result is a combination that traditional BPM struggles to achieve — high operational flexibility alongside maintained managerial control. This is the logic of the learning organisation: not merely reacting to environmental change, but building a lasting capacity to recognise, interpret, and exploit it [24].

The technological operationalisation of this logic points directly to Adaptive Case Management (ACM) as one of the earliest and most mature IT implementations consistent with dynamic BPM [14,20]. As postulated by Reichert and Weber [26], ACM supports flexibility in all phases of the process life cycle, including documentation of actual execution, near-the-point-of-action decision-making, and the treatment of successive process executions as the basis for organisational learning — operationalising what dynamic BPM articulated conceptually.

Answering RQ1: contemporary BPM in organisational practice already fulfills the logic of dynamic BPM. In accordance with, among others, According to the research of the Polish BPM Community, semi-structured and unstructured processes predominate and are of the highest importance for the organisation, for which the success of process execution is co-determined by the knowledge, engagement, and use of the dynamism of participants [6,12,22,27]. Dynamic BPM was not a marginal alternative — it was an earlier articulation of the direction BPM was always heading.

## 5    Discussion

To assess the scale of academic engagement with dynamic BPM, a bibliometric analysis was conducted in Scopus using the query TITLE-ABS-KEY ("dynamic business process management" OR "dynamic BPM"), covering all document types without language restrictions. The results are unambiguous: 38 publications were indexed between 2006 and 2026, of which 26 had been cited, generating 276 citations and an h-index of 10. For a concept developed over two decades, this does not merely indicate marginal-

isation, but near-invisibility. To contextualise this limited academic visibility, the results were compared with the publication output returned by two broader Scopus queries representing related research streams. The query TITLE-ABS-KEY ("agile processes" OR "agile BPM") returned 1,308 publications indexed between 1994 and 2026, (1,197 between 2006 and 2026) whereas TITLE-ABS-KEY ("flexibility in processes" OR "flexible processes") returned 1,851 publications indexed between 1971 and 2026 (1,496 between 2006 and 2026), with one additional record already assigned by Scopus to 2027. For comparability with the Dynamic BPM query and to ensure the readability of the figure, Fig. 2 presents annual publication counts for all three research streams only within the common observation period of 2006–2026.

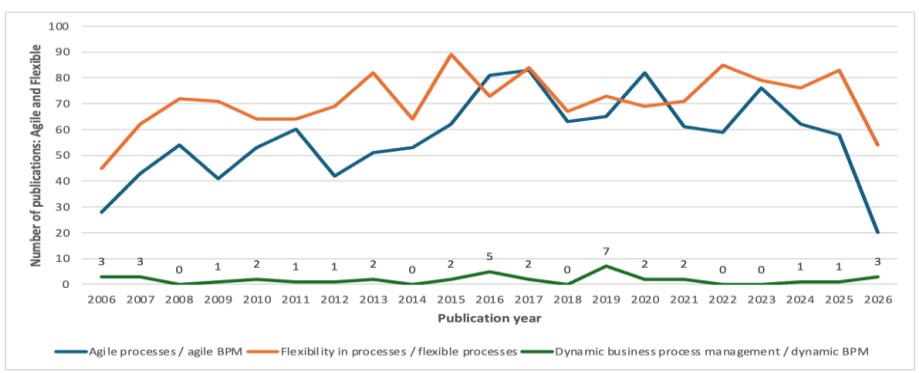

**Fig. 2** Number of publications indexed in Scopus between 2006 and 2026 retrieved using queries related to Dynamic BPM, Agile BPM, and process flexibility. *Source: Scopus.com, access: 02.06.2026.*

Yet the concept was not rejected. It was simply never seriously engaged. Four factors explain this:

**I. It came from practice, not theory.** Dynamic BPM emerged from direct observation of business failure — not from within the BPM research community. Like ACM, it was initially perceived as a practitioner workaround rather than a theoretical contribution. The problems it addressed — environmental instability, semi-structured processes, the limits of ex ante design — only began entering broader academic debate in the second decade of the 2000s [7,9].

**II. The field was focused elsewhere.** BPM development in the first two decades of this century was dominated by technologies supporting modelling, automation, and control: notations, workflow engines, document management, RPA, process mining [8,12,13,20,26]. In that context, concepts refining tools for standardised processes attracted more attention than concepts challenging the underlying logic of process management.

**III. It required a paradigm shift.** Adopting dynamic BPM meant moving from predictability, standardisation, and control towards unpredictability, execution context, participant agency, and organisational learning. Even in an innovative research environment, it was easier to treat dynamic BPM as a niche case than to accept the depth of

revision it implied. The publications of Kemsley, Olding and Rozwell, and Di Ciccio [7–9] eventually began to shift this — but slowly.

**IV. It demanded BPM–KM integration.** A full understanding of dynamic BPM directly leads to the necessity of integrating process management with knowledge management [1] — a step that for a long time placed it outside the dominant interests of BPM research, focused more on design, modelling, and control than on knowledge creation and operationalisation during execution. Only the rise of AI and knowledge-intensive processes is making this integration unavoidable [21,28].

In many organisations, BPM evolved away from the predefined-model paradigm towards dynamic, contextual, knowledge-driven management - fulfilling the logic of dynamic BPM without naming it. Answering RQ2: dynamic BPM was marginalised and forgotten, not rejected. Its limited presence in the literature reflects misalignment with dominant research interests of its time, not a verdict on its validity.

The analysis points to four areas where the further development of BPM requires extending the assumptions of dynamic BPM [1,11,29].

**AI as a process actor.** Hyperautomation and AI are introducing a new type of process participant: agents capable of interpreting context, initiating actions, recommending decisions, and executing tasks with partial autonomy [29,30]. This requires explicit recognition of AI agency within process management — defining authorisations, constraints, oversight principles, accountability, and conditions for human–AI collaboration — in the same way dynamic BPM defined these for human participants.

**Extended BPM governance.** Under conditions of growing AI autonomy, governance must expand to encompass limits of autonomous action, escalation principles, accountability for AI-assisted decisions, agent oversight, and the possibility of human intervention during execution [11,29]. Process modelling must similarly evolve to accommodate contextual variability, runtime reconfiguration, and shared human–AI decision-making.

**Deeper BPM–KM integration.** Knowledge must be treated as a dynamic process component — co-created, updated, and used by both humans and AI systems [29]. This extends and deepens the logic already present in dynamic BPM's third principle: that process execution and knowledge documentation are inseparable.

**Broader BPM objectives.** Beyond operational efficiency, BPM objectives must now encompass resilience, sustainable development, AI transparency, and the quality of human–technology collaboration [29,30] — restoring the balance between the technological and managerial dimensions of BPM that dynamic BPM originally sought to establish.

Answering RQ3: dynamic BPM does not provide a closed answer to the emerging stage of BPM evolution, but it provides an exceptionally apt conceptual foundation. Its core assumptions — diverse process nature, runtime adaptation, practical knowledge, and responsible autonomy — remain valid under hyperautomation and AI. What is needed is their extension to include AI governance, shared human–AI agency, a new architecture of accountability, and broader BPM objectives. The future development of BPM is best understood not as a departure from dynamic BPM, but as its next evolutionary stage.

## 6    Conclusion

Dynamic BPM was ahead of its time in two senses. Managerially, it shifted attention from the excellence of the ex ante optimised standard towards real-time execution, runtime adaptation, participant accountability, and organisational learning — before the mainstream of BPM research was ready to follow. Technologically, it anticipated the need for system classes like ACM that do not merely automate a predefined flow, but support its dynamic execution, documentation, and reinterpretation [14].

The fact that it did so without broader recognition is the central finding of this article — and a revealing one. Dynamic BPM was not a failed experiment. It was not reviewed and found wanting. It was simply not reviewed as a theoretical alternative. Its 34 Scopus-indexed publications over two decades, its near-absence from BPM handbooks and research agendas, reflect not a scholarly verdict but a structural blind spot: the field's sustained orientation towards predictability, standardisation, and technological control left little room for a concept whose value lay precisely in questioning those assumptions. What makes this worth revisiting now is not nostalgia. It is the recognition that the questions dynamic BPM raised — about process variability, participant agency, knowledge creation during execution, and the limits of ex ante design — are exactly the questions that hyperautomation and agentic AI are forcing back onto the BPM agenda. The concept requires extension: AI governance, shared human–AI agency, new accountability architectures, and broader organisational objectives beyond efficiency [29,30]. But the foundational logic holds. Managing processes means managing the conditions under which knowledge, judgement, and responsible autonomy can be effectively exercised — by humans, by AI agents, or by both working together. For the BPM research community, the lesson may be as much about the field itself as about dynamic BPM. Concepts that challenge dominant paradigms rather than refine them face structural disadvantages in academic discourse — harder to publish, harder to cite, harder to build communities around. Dynamic BPM is a case study in how a field can overlook what it most needs. Recovering it is not just an act of historical justice. It is a practical step towards a BPM theory adequate to the organisations that actually exist.

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
