# OpenReview forum: "The Rediscovery of Dynamic BPM: From Theoretical Margins to Mainstream Practice"
_bpm-conference.org/BPM/2026/Workshop/FOR-BPM — FOR-BPM_

### Official Review · Reviewer_T9PT · 2026-07-03
**The Rediscovery of Dynamic BPM: From Theoretical Margins to Mainstream Practice**

**Rating:** 3

**Review:**

This paper revisits the “forgotten” concept of Dynamic BPM. Based on a literature analysis, it argues that Dynamic BPM has long been a promising paradigm for managing business processes with greater flexibility than traditional BPM approaches that rely on ex ante process specifications. However, because the concept emerged primarily from practice rather than academia, it has received limited scholarly attention. The authors further argue that recent technological and organizational developments—particularly the emergence of agentic AI and the growing prevalence of knowledge-intensive processes—make it timely to reassess Dynamic BPM as an approach to business process management.

I generally agree with the authors’ central argument. Recent technological and organizational developments have indeed increased the need for more flexible and adaptive process execution support. Revisiting the Dynamic BPM concept and assessing its relevance within today’s BPM landscape is therefore both timely and worthwhile. The paper presents a clear and convincing analysis of why Dynamic BPM did not gain widespread adoption in the past and why its underlying ideas may be even more relevant today.

However, a major weakness of the paper is that the concept of Dynamic BPM itself remains insufficiently defined. The descriptions provided are rather abstract and there appears to be no core reference that explicitly develops Dynamic BPM as a distinct paradigm. In fact, only a single reference is mentioned that explicitly uses the term Dynamic BPM, and this publication dates from 2019 rather than from the early foundational period mentioned (2005/2006). As a result, it is difficult to understand precisely what constitutes Dynamic BPM, how it emerged as a distinct concept, and how it differs from existing approaches. To strengthen the paper, I would recommend:
1. Providing a more explicit definition and conceptualization of Dynamic BPM in the paper.
2. Including more references to Dynamic BPM, even if these originate from professional or grey literature rather than scientific publications, given the practice-based origin of the concept. The literature review revealed 34 paper related to Dynamic BPM. It would be good to provide transparency in this literature review process and at least include all references (possibly as an appendix).
3. Conducting a more in-depth comparison with established approaches to process flexibility, such as case handling, declarative process modeling, ADEPTflex, adaptive case management, and context-aware BPM. The paper should clearly articulate in which respects Dynamic BPM overlaps with and differs from these approaches. Useful references in this regard include:

     •	Cognini, R., Corradini, F., Gnesi, S., et al. (2018). Business process flexibility: A systematic literature review with a software systems perspective. Information Systems Frontiers, 20, 343–371. https://doi.org/10.1007/s10796-016-9678-2
     •	Karastoyanova, D., & Grefen, P. (2024). Non-standard BPM platforms. In Handbook on Business Process Management and Digital Transformation (pp. 138–168). Edward Elgar Publishing.

In addition, the paper would benefit from a stronger forward-looking perspective on the potential re-emergence of Dynamic BPM. While the authors make a compelling case for reconsidering the concept, they do not sufficiently discuss:
1. Why the current context makes this the right moment to revisit Dynamic BPM.
2. Why Dynamic BPM is particularly well suited to address the future directions of BPM evolution discussed in Section 5.2. Addressing this point also requires a clearer definition of the approach itself as already indicated in my remark above.
3. What concrete steps, research directions, or technological advancements would be necessary to support broader adoption and further development of Dynamic BPM.

Overall, the paper presents a compelling argument for revisiting a potentially valuable concept that has received insufficient attention over the past decades. The rationale for reconsidering Dynamic BPM is well developed. However, the paper could be significantly strengthened by a more rigorous conceptualization of Dynamic BPM, a clearer positioning relative to existing flexibility-oriented BPM approaches, and a more explicit discussion of its future viability and adoption path. These additions would make the proposal more convincing and impactful and would certainly increase the discussion potential of the paper during the workshop.

**Advancing Bpm Thinking:**

Please see my suggestions in the review above. In summary:
- the conceptualization of Dynamic BPM could be improved in the paper so that every reader understands it in enough detail.
- the similarities and differences between Dynamic BPM and existing flexible BPM approaches should be compared to understand better what makes Dynamic BPM stand out, or suitable for the new context with technological and organizational evolutions
- a more explicit roadmap/discussion should be included of what is needed for broader adoption and further development of the Dynamic BPM paradigm this time. Which gaps are there? How can they be filled? et cetera.

---

### Official Review · Reviewer_7MNM · 2026-07-06
**Interesting and timely discussion on the rediscovery of dynamic BPM, but the central argument requires stronger positioning and evidence.**

**Rating:** 4

**Review:**

This paper is well aligned with the goals of the FOR-BPM workshop. Revisiting dynamic BPM as a potentially overlooked research direction is an interesting idea that is likely to stimulate discussion within the BPM community, particularly in light of current developments in hyperautomation and agentic AI. The paper is generally well written and presents a clear narrative around the authors' central claim.
Quality. The paper is thoughtfully written and develops a coherent line of argument. However, I found the evidence supporting its central thesis less convincing than I initially expected. In particular, the literature review appears to be limited primarily to publications explicitly using the term "dynamic BPM". This narrow focus overlooks several closely related research streams that have addressed similar challenges from different perspectives, such as knowledge-intensive processes, process flexibility, and workarounds, among others. As a consequence, the paper does not sufficiently demonstrate that the underlying ideas were genuinely forgotten rather than developed under different terminology. Moreover, the review reads more as a collection of references than as a critical analysis supporting the paper's claims. Some important statements would also benefit from stronger evidence and more complete referencing. For example, the paper relies on a recent study to argue for the growing importance of non-structured processes without sufficiently discussing the scope and representativeness of the study. Likewise, several statements referring to broader management literature (e.g., regarding the "hidden factory" and informal workarounds) should be supported with appropriate references.
Clarity. The paper is clearly structured and easy to follow. The motivation, historical narrative, and the proposed future directions are presented in a logical way. I felt, however, that the tone occasionally becomes somewhat defensive regarding the historical reception of dynamic BPM. While it is perfectly legitimate to argue that a research direction has been overlooked, presenting this argument in a more balanced and evidence-driven manner would make the paper more persuasive.
Originality. Re-examining dynamic BPM through the perspective of forgotten or overlooked research is an original and timely contribution that fits well with the objectives of FOR-BPM. The attempt to relate this discussion to current developments in AI and hyperautomation provides an interesting perspective. At the same time, I believe the originality claim would be strengthened by a broader engagement with adjacent research areas that have addressed similar problems under different terminology.
Significance. I believe the paper has the potential to stimulate valuable discussion within the BPM community. It raises interesting questions about how research ideas evolve, why some concepts disappear from mainstream discourse, and whether current developments provide an opportunity to revisit them. Even though I am not fully convinced by all aspects of the historical argument, I think the paper can foster a worthwhile discussion about dynamic BPM, its relationship to neighbouring research streams, and its relevance in the context of AI-enabled BPM.
Pros
•	Excellent fit with the goals and spirit of the FOR-BPM workshop.
•	Timely and thought-provoking topic connecting historical BPM research with current developments in AI and hyperautomation.
•	Clear structure and generally well-written presentation.
•	Provides a constructive discussion of emerging research directions.
Cons
•	The literature review is too narrowly focused on the explicit term "dynamic BPM" and does not sufficiently engage with closely related research streams.
•	The central claim that dynamic BPM was genuinely overlooked is not yet fully substantiated.
•	Several arguments would benefit from stronger evidence and more complete referencing.
•	A more balanced tone would make the paper's central message more persuasive.

**Advancing Bpm Thinking:**

This contribution has the potential to advance the BPM community's thinking by encouraging reflection on whether important research ideas have been overlooked, renamed, or absorbed into other research streams over time. To maximise this contribution, I encourage the authors to broaden the discussion beyond publications explicitly using the term "dynamic BPM" and engage more deeply with related areas that have addressed similar challenges under different concepts, such as knowledge-intensive processes, process flexibility, and workarounds. This would help clarify what is genuinely distinctive about dynamic BPM and how it relates to the broader evolution of BPM research.
The paper could further strengthen its impact by providing more robust evidence for its historical and contemporary claims, particularly when arguing that current BPM practice has implicitly adopted the principles of dynamic BPM. Clarifying the scope and limitations of the empirical evidence cited and supporting broader statements with appropriate references would improve the credibility of the argument.
The section on emerging directions of BPM evolution is a valuable addition and provides a constructive perspective on how dynamic BPM can inform future research in areas such as AI, governance, and knowledge management. In my view, these forward-looking ideas would become even more compelling if they were grounded in a stronger analysis of the historical literature and a broader engagement with related BPM research. This would reinforce the paper's central claim and make the connection between the "forgotten" framework and its proposed future evolution more convincing.

---

### Official Review · Reviewer_WRSD · 2026-07-06
**The paper picks up the topic of "dynamic BPM" and discusses why this is a previously neglected, but exciting and relevant research topic. For this, a literature review is conducted, followed by defining the essence and basic principles of dynamic BPM.**

**Rating:** 3

**Review:**

The paper picks up the topic of "dynamic BPM" and discusses why this is a previously neglected, but exciting and relevant research topic.
For this, a literature review is conducted, followed by defining the essence and basic principles of dynamic BPM.

Strenghts:
- The paper is definitely a very good fit for the workshop's goals.
- The paper is interesting to read: the reasons why dynamic BPM was neglected in the first place are interesting and convincing and might provide a good basis for discussion in the workshop.
- The paper manages to raise interest to re-visit the topic of dynamic BPM. It convincingly explains the wide applicability and with this the relevance of the topic.
- The emerging directions are interesting are on point with current developments and have also the potential to spark discussions at the workshop.

Action points:
- I would abstain from referring to a "systematic" literature review. The paper misses out many related works. Please try to search for "dynamic processes" and there will be many more papers. I guess this is because the paper is taking a management perspective on BPM rather than a more technical one, but lacking a discussion of a whole line of research is limiting the overall message of the paper, i.e., dynamic BPM as neglected topic.
See, for example, the reflections by Wil van der Aalst at BPM 2012: A Decade of Business Process Management Conferences: Personal Reflections on a Developing Discipline: several of the mentioned use cases refer to dynamic processes. Also see Manfred Reichert, Barbara Weber:
Enabling Flexibility in Process-Aware Information Systems - Challenges, Methods, Technologies. Springer 2012, ISBN 978-3-642-30408-8, pp. I-XVIII, 1-515
- What is the connection or difference between dynamic BPM and flexible processes?
This need a more elaborated discussion.
- The directions are interesting. A bit more elaboration on what this means for research in terms of one or two research directions would be interesting.

**Advancing Bpm Thinking:**

- A better distinctionn of "dynamic BPM", "dynamic processes", "flexibility in processes", etc is required.